# An EKF-Based Method and Experimental Study for Small Leakage Detection and Location in Natural Gas Pipelines

**Qingmin Hou [1] and Weihang Zhu [2],\***

[1] School of Energy and Building Engineering, Harbin University of Commerce, Harbin 150028, China
[2] Department of Engineering Technology, College of Technology, University of Houston, Houston, TX 77204, USA
\* Correspondence: wzhu21@central.uh.edu; Tel.: +86-1-713-743-6980

**Abstract:** Small leaks in natural gas pipelines are hard to detect, and there are few studies on this problem in the literature. In this paper, a method based on the extended Kalman filter (EKF) is proposed to detect and locate small leaks in natural gas pipelines. First, the method of a characteristic line is used to establish a discrete model of transient pipeline flow. At the same time, according to the basic idea of EKF, a leakage rate is distributed to each segment of the discrete model to obtain a model with virtual multi-point leakage. As such, the virtual leakage rate becomes a component of the state variables in the model. Secondly, system noise and measurement noise are considered, and the optimal hydraulic factors such as leakage rate are estimated using EKF. Finally, by using the idea of an equivalent pipeline, the actual leakage rate is calculated and the location of leakage on the pipeline is assessed. Simulation and experimental results show that this method can consistently predict the leakage rate and location and is sensitive to small leakages in a natural gas pipeline.

**Keywords:** natural gas pipeline; small leaks; pipeline leakage detection; extended Kalman filter

## 1. Introduction

Natural gas is a kind of high-quality, efficient, clean energy and raw material. Since the 1970s, the development of natural gas in the world has been accelerated, and the research on gas exploitation, transportation, and storage has received increasing attention [1–7]. Nowadays, natural gas pipelines account for almost half of the total length of pipelines in the world. Natural gas is poisonous and explosive. Therefore, it is important that natural gas pipelines are maintained in a safe manner. Leakages are the major causes of in-service natural gas pipeline accidents. In practice, natural gas pipeline leakages can be divided into two types—sudden and rapid leakages with a high leakage rate, and slow leakages with a low leakage rate, also known as seepage.

Sudden leakages are mostly caused by third-party earth work, heavy vehicle rolling, geological changes, and impacts [8,9]. There is extensive literature on sudden leakage detection [10–13], such as the commonly used negative pressure wave (NPW) method [14–16] and the fiber-optic-based method [17,18]. The principle of NPW is that when leaks develop in a natural gas pipeline, the gas density near the leak point will decrease rapidly. This phenomenon results in a negative pressure wave which propagates through the gas from the leak point. The NPW method often employs pressure sensors or other types of sensors that can detect wave propagation. Thanks to the recent development in structural health monitoring [19–26] and damage detection [27–33], many new types of sensors, such as piezoceramic transducers [34–39] and fiber optic sensors [40–43], have been developed for civil infrastructure. Because of their high bandwidth and low cost, piezoceramic transducers have

been applied to pipeline monitoring [44–46]. On the other hand, because of their distributive nature and small size, fiber optic sensors are often reported for pipeline monitoring [47–49]. In the NPW method, these sensors are installed upstream and downstream of a pipeline to detect negative pressure waves [50–54]. The signal captured by these pressure sensors can be processed to determine whether there is a gas leak. Fiber-optic-based methods usually use distributed fiber optical sensors to detect leakages in natural gas pipelines; a distributed fiber optical sensor installed along the pipeline can detect the temperature variation or vibrations caused by leakage. The commonality of these two methods is the utilization of sensors. This leads to a potential problem that if the sensors are not accurate enough, it is highly possible to miss the alarms for small leakages.

Slow small leakages (seepage) usually occur in weak links of the pipelines, such as welding seams, flanges, or valves [55–57], and they are difficult to detect due to their concealment. In addition, pipelines are subject to corrosion [58–60], erosion [61], and cracking [62–64], which may also cause small leakages at the initial stage. Currently, there are only limited studies in this area [65,66]. At present, the effective method for small leakages is based on the real-time model method. Compared with other leakage detection methods, the real-time model method can provide all the characteristics of pipeline flow and fluid properties. It also considers the impact of physical properties of pipelines. In addition, the real-time model considers the influence of system noise and measurement noise and has advantages in small leakage detection. However, it also has some disadvantages, such as complex computation and slow convergence.

The Kalman filter is a kind of filtering algorithm that estimates the required states through algorithms from the observation quantity related to the extracted signal, however, it can only be used for linear systems. The Kalman filter that can estimate states for a nonlinear system is called the extended Kalman filter (EKF) [67]. Continuous nonlinear equations need to be linearized first when using EKF. The EKF has the advantages of fast convergence and high accuracy in state estimation [68–71]. This paper proposes an EKF-based method for small leakage detection and location in natural gas pipelines.

Most researchers working in this topic approach the leakage detection problem from the research field of control. The main limitation in the extant literature is about the applicability of different control and detection methods. Complementary to the previous efforts, the authors of this paper approach this topic as subject matter experts in pipe leakage detection. We understand that both the NPW method and the fiber-optic-based method are widely used and able to obtain reasonably good leakage detection results. These two methods do not need to establish a pipeline model. They mainly process signals and identify patterns for real-time leakage detection. However, these two methods mainly work well with the common sudden and rapid leakage detection. They perform poorly with the slow and small leakage. EKF-based method is essentially a real-time modeling method. This method requires pipeline models. The computation is complex and converges slowly. Therefore, this method is not suitable for sudden and rapid leakage detection. On the other hand, a real-time model can provide a pipeline's flow characteristics and fluid properties and consider pipeline physical properties as well. In addition, a real-time model considers the effect of system noise and measurement noise. Its working principle determines its advantage in slow and small leakage detection. The Kalman filter has the advantages of fast convergence and high precision in terms of state estimation. This compensates for the original shortcomings in the real-time model method such as slow convergence. Therefore, it is adopted for real-time leakage detection in this paper.

The rest of the paper is organized as follows. Section 2 proposes the pipeline transient flow model containing the virtual multi-point leakages. Section 3 elaborates the mathematics for the state estimation by EKF. Section 4 explains the calculation of the actual leakage rate and location. Section 5 verifies the model with a simulation example. In Section 6, a physical experiment is conducted to validate the effectiveness of the proposed method. Section 7 concludes the paper.

## 2. The Pipeline Transient Flow Model Containing the Virtual Multi-Point Leakages

Since EKF can only be applied to the form of discrete time and discrete space for state estimation, it is necessary to discretize the pipeline length and leakage rate. The basic idea is presented below. First, the pipeline model is divided into $N-1$ segments, and the position coordinates on each segment point are $x_1, x_2, \ldots, x_i, x_{i+1}, \ldots, x_{N-1}, x_N$. Assume the leakage happens at points $x_2, \ldots, x_i, x_{i+1}, \ldots, x_{N-1}$, $N-2$ segments. At time $t$, the pressure, flow, and virtual leakage rate at point $x_i$ are represented as $P(x_i, t)$, $Q(x_i, t)$, and $K(x_i, t)$, respectively. The actual leaking pipeline and the virtual multi-point leaking pipeline are shown in Figures 1 and 2, respectively.

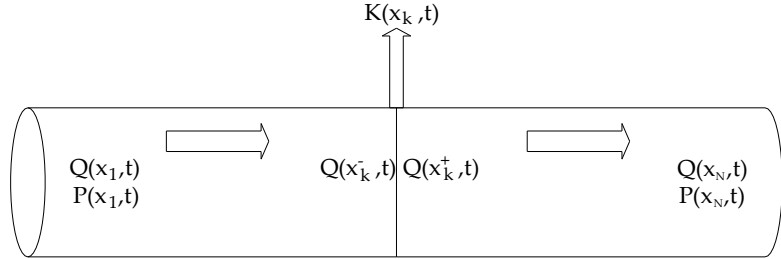

**Figure 1.** A real leakage in a continuous pipeline model.

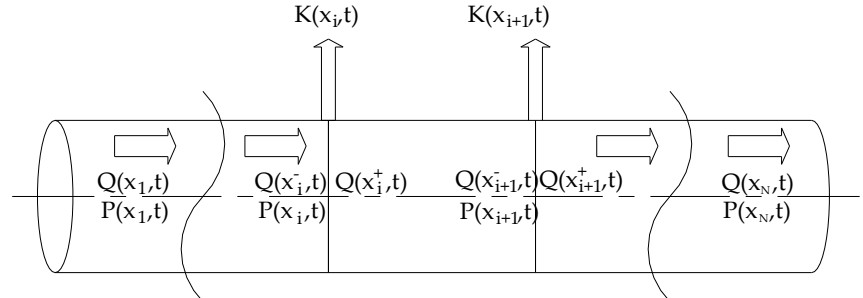

**Figure 2.** Virtual multi-point leakages in a discrete pipeline model.

The method of a characteristic line is used to solve the basic control equation of natural gas flow. Combining the above ideas, the leakage is virtually distributed at segment points, as shown in Figure 3, and the corresponding discrete equations can be obtained as [72,73]

$$(P_{i,j}^- - P_{i-1,j-1}^+) + \frac{c}{A}(Q_{i,j}^- - Q_{i-1,j-1}^+) + \frac{\lambda c^3 \Delta t}{4DA^2}\left(\frac{Q_{i,j}^-|Q_{i,j}^-|}{P_{i,j}^-} + \frac{Q_{i-1,j-1}^+|Q_{i-1,j-1}^+|}{P_{i-1,j-1}^+}\right) = 0 \tag{1}$$

$$(P_{i,j}^+ - P_{i+1,j-1}^-) - \frac{c}{A}(Q_{i,j}^+ - Q_{i+1,j-1}^-) - \frac{\lambda c^3 \Delta t}{4DA^2}\left(\frac{Q_{i+1,j-1}^-|Q_{i+1,j-1}^-|}{P_{i+1,j-1}^-} + \frac{Q_{i,j}^+|Q_{i,j}^+|}{P_{i,j}^+}\right) = 0 \tag{2}$$

where $i$ is the subscript to represent a segment $x_i$, $j$ is the subscript to represent time $t_j$, $Q_{i,j}^-$ is mass flow before the leakage point at the segment $x_i$ at time $t_j$, kg/s, $Q_{i,j}^+$ is mass flow after the leakage point, kg/s, $P_{i,j}^-$ is pressure before the leakage point, Pa, $P_{i,j}^+$ is pressure after the leakage point, Pa, $A$ is the cross-sectional area of pipe, m$^2$, $c$ is the speed of sound, m/s, $D$ is pipe diameter, m, and $\lambda$ is the hydraulic friction coefficient.

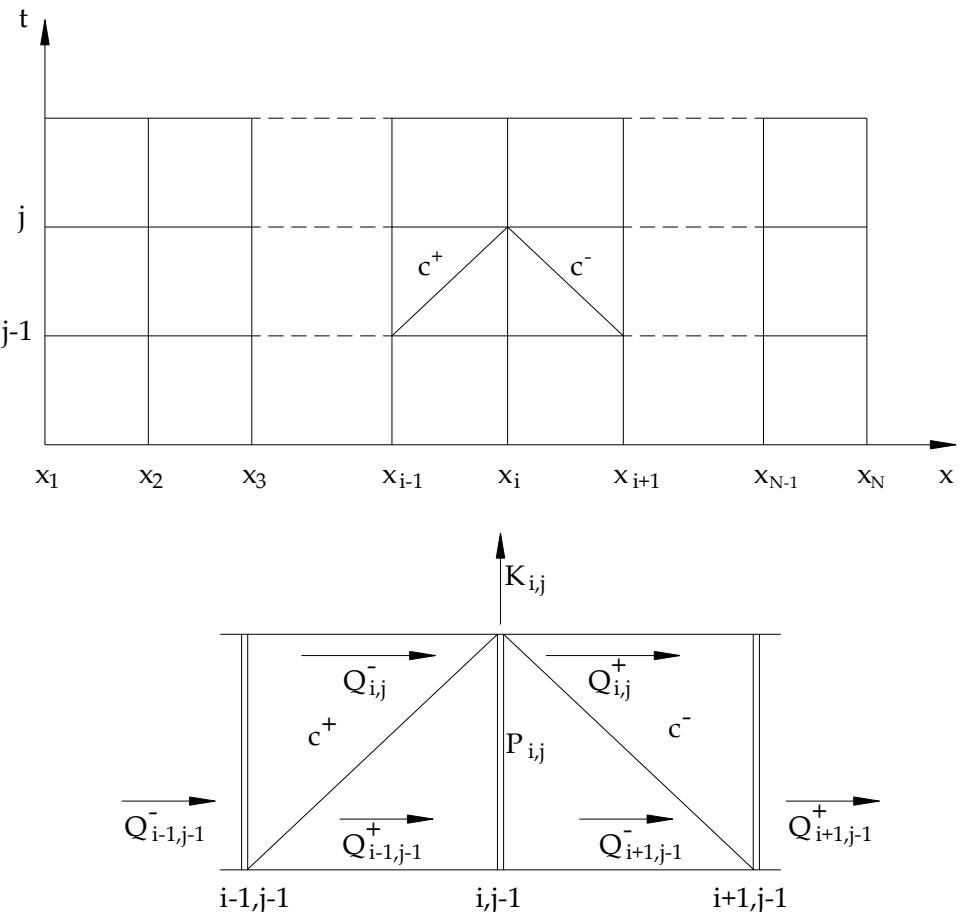

**Figure 3.** Discretization for virtual multi-point leakage in a pipeline.

Since the leakage occurs in the vertical direction, it can be assumed that the momentum change caused by the leakage in the horizontal direction can be ignored. According to the principle of mass conservation, the boundary conditions at the leakage point can be given as

$$P_{i,j}^+ = P_{i,j}^- = P_{i,j} \tag{3}$$

$$Q_{i,j}^- - Q_{i,j}^+ = K_{i,j}. \tag{4}$$

Among them, $i = 1, 2, \ldots, N; j = 1, 2, \ldots, P_{i,j}^-$, and $Q_{i,j}^-$ represent the mass flow and pressure at time $t_j$ at $x_i^-$ position (before the leakage point), namely $P_{i,j}^- = P(x_i^-, t_j)$ and $Q_{i,j}^- = Q(x_i^-, t_j)$. In the same way, $P_{i,j}^+ = P(x_i^+, t_j), Q_{i,j}^+ = Q(x_i^+, t_j), P_{i,j} = P(x_i, t_j), K_{i,j} = K(x_i, t_j)$, and $x_i = (i-1)\Delta x$.

The EKF-based method proposed in this paper assumes that the leakage rate at each segment point is constant in time [74–78], namely,

$$K_{i,j} = K_{i,j-1}. \tag{5}$$

Equations (1)–(5) are based on the basic idea of filter estimation, and the discretized pipeline transient flow model including virtual multi-point leakage can now be established, as follows.

To make the virtual leakage rate a component of the state variables in the above model, Equations (3) and (4) are substituted into Equations (1) and (2) to give

$$(P_{i,j} - P_{i-1,j-1}) \quad + \frac{c}{A}(Q_{i,j}^- - Q_{i-1,j-1}^- + K_{i-1,j-1}) + \frac{\lambda c^3 \Delta t}{4DA^2}$$
$$\times (\frac{Q_{i,j}^-|Q_{i,j}^-|}{P_{i,j}} + \frac{(Q_{i-1,j-1}^- - K_{i-1,j-1})|Q_{i-1,j-1}^- - K_{i-1,j-1}|}{P_{i-1,j-1}}) = 0 \tag{6}$$

$$(P_{i,j} - P_{i+1,j-1}) \qquad -\frac{c}{A}(Q^-_{i,j} - K_{i,j} - Q^-_{i+1,j-1}) - \frac{\lambda c^3 \Delta t}{4DA^2}$$
$$\times \left( \frac{Q^-_{i+1,j-1}\left|Q^-_{i+1,j-1}\right|}{P_{i+1,j-1}} + \frac{(Q^-_{i,j} - K_{i,j})\left|Q^-_{i,j} - K_{i,j}\right|}{P_{i,j}} \right) = 0 \tag{7}$$

Equations (5)–(7) constitute a nonlinear implicit system with the following variables: pressure at each segment point $P_{i,j}(i = 1, \ldots, N)$, the flow rate to the left of each segment point $Q^-_{i,j}(i = 1, \ldots, N)$, and the virtual leakage rate at each segment point. These variables constitute a dimensional state vector:

$$X_j = [P_{1,j}, P_{2,j}, P_{3,j}, \cdots, P_{N,j}, Q^-_{1,j}, Q^-_{2,j}, \cdots, Q^-_{N-1,j}, Q^-_{N,j}, K_{2,j}, K_{3,j}, \cdots, K_{N-1,j}]^T.$$

Therefore, the nonlinear implicit system can be expressed as

$$F(X_j, X_{j-1}) = 0 \tag{8}$$

$$\begin{cases} P_{1,j} = f_P(t) \\ Q^-_{N,j} = f_q(t) \end{cases} \tag{9}$$

where $f_P(t)$ and $f_q(t)$ is the boundary condition of $P$ and $Q$ in this system.

## 3. State Estimation by EKF

To use the EKF method, the system needs to be linearized first. Assuming that the optimal estimate of $X_j$ is $X_s$, the corresponding state Equation can be written according to the EKF,

$$x_j = A_s x_{j-1} + w_{j-1} \tag{10}$$

where $x_j = X_j - X_s$, $A_s$ is the transition matrix that can be obtained in the linearization process, and $w_j$ is the system noise. The corresponding measurement Equation can be written as

$$z_j = H x_j + v_j \tag{11}$$

where $H$ is the measurement matrix and $v_j$ is the measurement noise. Both system noise and measurement noise are assumed to be zero-mean Gaussian white noise in the Kalman filter.

Let $\overline{X}_j = X_{j/j-1}$ represent the prior estimate of the state variables at time $j$ obtained using the optimal estimate at time $j-1$. Let $\hat{X}_j = X_{j/j}$ represent the optimal estimate of the state variables at time $j$. Correspondingly:

$$\overline{x}_j = \overline{X}_j - X_s, \quad \hat{x}_j = \hat{X}_j - X_s. \tag{12}$$

In the following discussion, $\overline{P}_j$ is the variance matrix of the prior estimation error, $\hat{P}_j$ is the optimal estimation variance matrix ($\hat{P}_0$ is known), $L_j$ is the Kalman gain, and $R$ is the variance matrix of the measurement noise. The steps of applying EKF to estimate state variables are as follows [79,80]:

1.  With the initial value $(\hat{X}_0, \hat{P}_0)$, linearize the Equations near $\hat{X}_0$ and obtain the transfer matrix $A_s$;
2.  Solve the nonlinear Equations (5)–(7), as $F(X_j, X_{j-1}) = 0$ $P_{i,j} = P(x_i, t_j)$, and $\overline{X}_j$ can be computed and obtained: $\overline{x}_j = \overline{X}_j - X_s$; $\overline{P}_j = A_s \overline{P}_{j-1} A_s^T + Q$;
3.  Use the following formula to compute the Kalman gain: $L_j = \overline{P}_j H^T [H \overline{P}_j H^T + R]^{-1}$;
4.  Obtain the optimal estimate $\hat{X}_j$ by measuring the value $z_j$

$$\hat{x}_j = \overline{x}_j + L_j(z_j - H x_j), \quad \hat{X}_j = X_s + \hat{x}_j,$$

obtain the optimal estimated variance matrix $\hat{P}_j$ at time $j$

$$\hat{P}_j = \overline{P}_j - L_j H \overline{P}_j$$

obtain the optimal estimate $\hat{X}_j$ by measuring the value $z_j$

$$\hat{x}_j = \bar{x}_j + L_j(z_j - Hx_j), \ \hat{X}_j = X_s + \hat{x}_j,$$

and obtain the optimal estimated variance matrix $\hat{P}_j$ at time $j$

$$\hat{P}_j = \bar{P}_j - L_j H \bar{P}_j;$$

5.  Go back to step 2 and calculate the optimal estimate of the moment $j + 1$. From the above computation, we can obtain the optimal estimation of the state vector $X_j$.

## 4. Calculating the Actual Leakage Rate and Location

The above estimated state vectors $X_j$ include the pressure at each segment point $P_{i,j}(i = 1, \ldots, N)$, the flow to the left of each segment point $Q_{i,j}^-(i = 1, \ldots, N)$, and the virtual leakage rate $K_{i,j}(i = 2, \ldots, N-1)$ at each segment point. To obtain the real leakage rate and the location, the idea of an equivalent system is used. That is, the virtual multi-point leakage pipeline should have the same initial conditions and boundary conditions as the actual pipeline. The pipeline flow loss and pressure drop caused by both of them should be the same. The computation formula of the actual leakage rate and leakage location of the actual pipeline [73] can be defined as follows.

$$\begin{cases} \hat{K}_j = \sum\limits_{i=2}^{N-1} \hat{K}_{i,j} \\ \hat{x}_k = \dfrac{\sum\limits_{i=2}^{N-1} \hat{K}_{i,j} \cdot x_{ki}}{\hat{K}_j} \end{cases} \tag{13}$$

Using the virtual leakage rate at each segment point $\hat{K}_{i,j}$ and the known $x_{ki}$, combined with Equation (13), the actual leakage rate and leakage location of the actual pipeline can be obtained.

## 5. A Simulation Example

The model is first verified with a simulation example. In this example, the process noise and measurement noise of this simulated gas pipeline are assumed to be Gaussian white noise. The specific parameters are:

$$L = 10 \text{ km}, \ D = 0.25 \text{ m}, \ c = 330 \text{ m/s}, \ \lambda = 0.05.$$

The boundary conditions are: $P(0,t) = 2$ MPa, $Q(L,t) = 2.6$ kg/s.

Generally, the length of each segment needs to be slightly larger than or equal to the distance that sound travels within one sampling period of the sensor ( ). If the segment length is shorter than that distance, then it is possible to miss the data. If the segment length is too long, then the precision may be reduced. In this example, the sampling frequencies $f$ of pressure sensors are assumed to be 1 Hz, therefore $\Delta x = c/f = 330$ m. Therefore, the pipeline is divided into $N - 1 = L/\Delta x \approx 30$ segments by the characteristic line method, namely $N = 31$.

It is assumed that the leakage occurs at 500 s, and the leakage point is 4 km away from the starting pressure sensor. To simulate a slow small leakage, set the leakage rate to be 4% of the total flow rate, i.e., 0.1 kg/s.

In this simulation example, there are three pressure observation points at 3.3 km, 6.6 km, and 9.9 km, namely $P_{11,j}$, $P_{21,j}$, and $P_{31,j}$, respectively.

The state vector is:

$$X_j = [P_{1,j}, P_{2,j}, P_{3,j}, \ldots, P_{31,j}, Q_{1,j}^-, Q_{2,j}^-, \ldots, Q_{31,j}^-, K_{2,j}, K_{3,j}, \ldots, K_{30,j}]^T.$$

Therefore, in this simulation example, the measurement matrix is a matrix of $3 \times 91$ $H(1, 11) = 1$, $H(2, 21) = 1$, and $H(3, 31) = 1$. The rest of the elements are set as 0.

The initial condition is:

$$\hat{X}_0 = [P_{1,0}, P_{2,0}, P_{3,0}, \cdots, P_{31,0}, Q^-_{1,0}, Q^-_{2,0}, \cdots, Q^-_{31,0}, K_{2,0}, K_{3,0}, \cdots, K_{30,0}]^T.$$

Among them:

$$P_{11,0} = 1,949,679 \text{ Pa}, \ P_{21,0} = 1,898,024 \text{ Pa}, \ P_{31,0} = 1,844,923 \text{ Pa}$$

$$Q^-_{1,0} = \cdots = Q^-_{31,0} = 2.6 \text{kg/s}$$

$$K_{2,0} = K_{2,0} = \cdots = K_{30,0} = 0$$

$$\hat{P}_0 = \begin{bmatrix} 0 & 0 & 0 & 0 & 0 \\ 0 & \sigma^2_{P0} I_{30} & 0 & 0 & 0 \\ 0 & 0 & \sigma^2_{Q0} I_{30} & 0 & 0 \\ 0 & 0 & 0 & 0 & 0 \\ 0 & 0 & 0 & 0 & \sigma^2_{K0} I_{29} \end{bmatrix}.$$

$\hat{P}_0$ is a $91 \times 91$ diagonal matrix, where $I_k$ is a $k \times k$ diagonal matrix. Here, the assumptions are:

$$\sigma^2_{P0} = 10^8 (\text{Pa})^2, \ \sigma^2_{Q0} = 0.3 (\text{kg/s})^2, \text{ and } \sigma^2_{K0} = 0.2 (\text{kg/s})^2.$$

The covariance of process noise and measurement noise is known. It is assumed that it does not vary with the state variable [12].

$$Q = \begin{bmatrix} 0 & 0 & 0 & 0 & 0 \\ 0 & \sigma^2_P I_{30} & 0 & 0 & 0 \\ 0 & 0 & \sigma^2_Q I_{30} & 0 & 0 \\ 0 & 0 & 0 & 0 & 0 \\ 0 & 0 & 0 & 0 & \sigma^2_K I_{29} \end{bmatrix}, \ R = \begin{bmatrix} \sigma^2_{PM} & 0 & 0 \\ 0 & \sigma^2_{PM} & 0 \\ 0 & 0 & \sigma^2_{PM} \end{bmatrix}$$

where $\sigma^2_P$ is the noise variance of pressure, $\sigma^2_Q$ and $\sigma^2_K$ are the variance of the noise of flow and leakage rate, respectively. $\sigma^2_{PM}$ is the measurement noise variance of pressure. Here, the assumptions are: $\sigma^2_P = 10^8 (\text{Pa})^2$, $\sigma^2_Q = 0.25 (\text{kg/s})^2$, and $\sigma^2_K = 0.1 (\text{kg/s})^2$.

Based on the steady pressure distribution along the pipeline, the steady state pressure values at the three pressure measurement points before and after leakage are calculated, and the Gaussian white noise is added to construct the pressure simulation data at the pressure measurement points, as shown in Figure 4.

By using the simulation data shown in Figure 4 as the measurement data, and using the iterative procedure based on the calculation steps described above, the estimated leakage rate and leakage location by EKF are obtained, as shown in Figures 5 and 6.

As can be seen in Figure 5, the design leakage rate has a step-wise change in flow rate representing a leak that commences at 500 s into the simulation, and the leakage rate is 4% of flow, i.e., 0.1 kg/s. It can be seen that the estimated leakage rate curve is basically consistent with the designed leakage rate curve, indicating that the proposed method based on EKF can detect the leakage and estimate the leakage rate accurately.

It is clear from Figure 6 that the design leakage location is 4 km downstream in the pipeline, and the curve is the estimated leakage location of the EKF. It can be seen from the figure that the two curves are also basically consistent, and the results show that the proposed method can locate the leakage position.

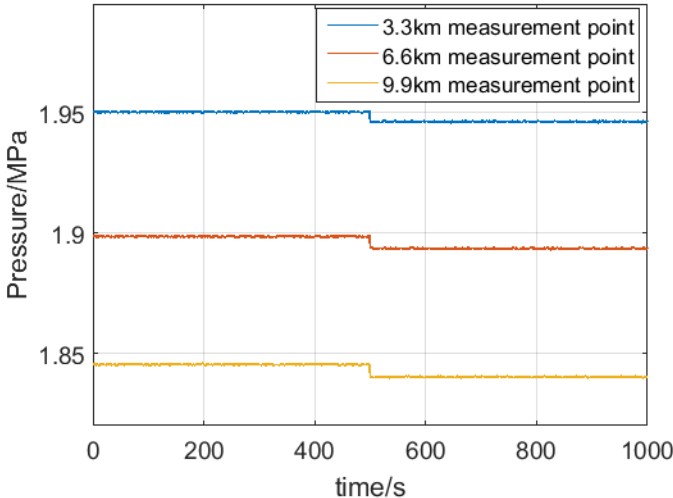

**Figure 4.** Pressure simulation data.

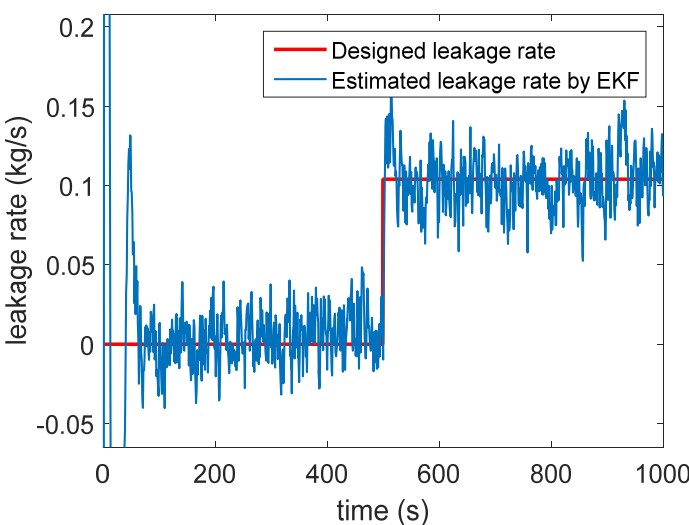

**Figure 5.** Estimated leak rate using simulation data. EKF: extended Kalman filter.

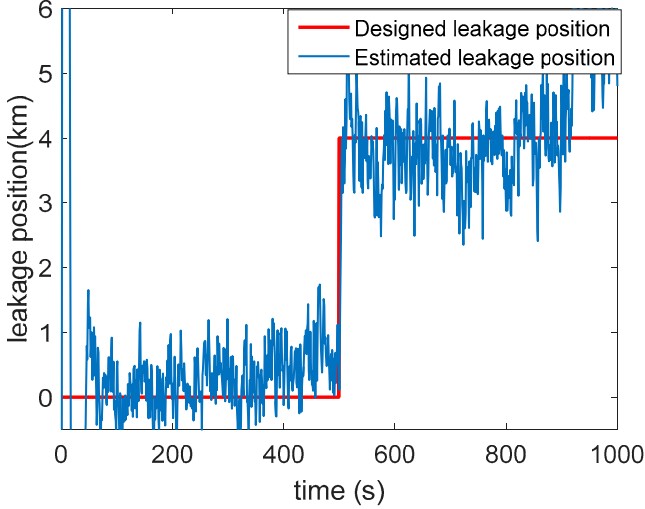

**Figure 6.** Estimated leak location using simulation data.

## 6. A Physical Experiment Case Study

Due to the flammable and explosive nature of natural gas, it is nearly impossible to carry out leakage rate measurement experiments using real natural gas in a pipeline. In this paper, an experimental platform was built, and compressed air was used instead of natural gas in the experiment for safety reasons.

The system setup, displaying the layout of the gas leakage experimental platform, is shown in Figure 7. The platform is mainly composed of pipeline sections, a compressor, and air tanks. The compressor is a belt piston machine, with a 7.5 kW rated power, 800 L/min displacement, and 1.25 MPa working pressure. The dimensions of the cylindrical air tanks are 1 × 2.4 m (diameter × height). The weight of the air tank is 515 kg. There is one location where a leak can be simulated, named leak point in Figure 7. A leakage can be simulated by opening valve V4, and a flowmeter is set to measure the leakage rate. There are two pressure sensors (P1, P2). The specific parameters of the experimental pipeline are:

$L$ (the length between P1 and P2) = 9 m, $D$ (inner diameter) = 0.253 m, $c$ = 300 m/s.

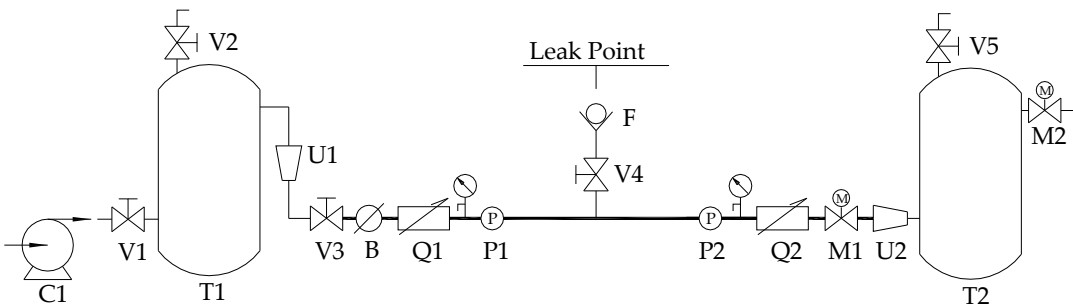

C—air compressor; T1, T2—air tank; P1, P2—pressure sensor; Q1, Q2—vortex precession flowmeter; B—butterfly valve; V1~V7—ball valve; M1, M2—solenoid valve; F—flowmeter; U1, U2—pressure-regulating valve

**Figure 7.** The system diagram of the gas pipeline leak detection experiment testbed.

The photos of the platform are shown in Figure 8.
The sampling frequencies $f$ of pressure sensors and flow sensors are 1000 Hz, so:

$$\Delta x = c/f = 0.3 \text{ m}.$$

The pipe is divided into $N - 1 = L/\Delta x = 30$ segments, where $N = 31$.
Referring to Figure 7, the specific steps of the experiment are as follows.

1. In order to simulate the actual natural gas pipelines, an air compressor is used to pressurize tank T1.
2. Once a stable pressure is achieved, the valve M1 is opened to form the flow from T1 to T2.
3. Adjusting the pressure of the gas storage tank and the opening of the butterfly valve B to maintain starting and ending pressures in the pipeline to be 800 and 750 kPa, respectively. At this point, the flow rate at the beginning of the pipeline is 132 kg/h and all the sensors begin to collect data.
4. After 80 s, the ball valve V4 is open to a certain degree to allow a leak at leak point 1.

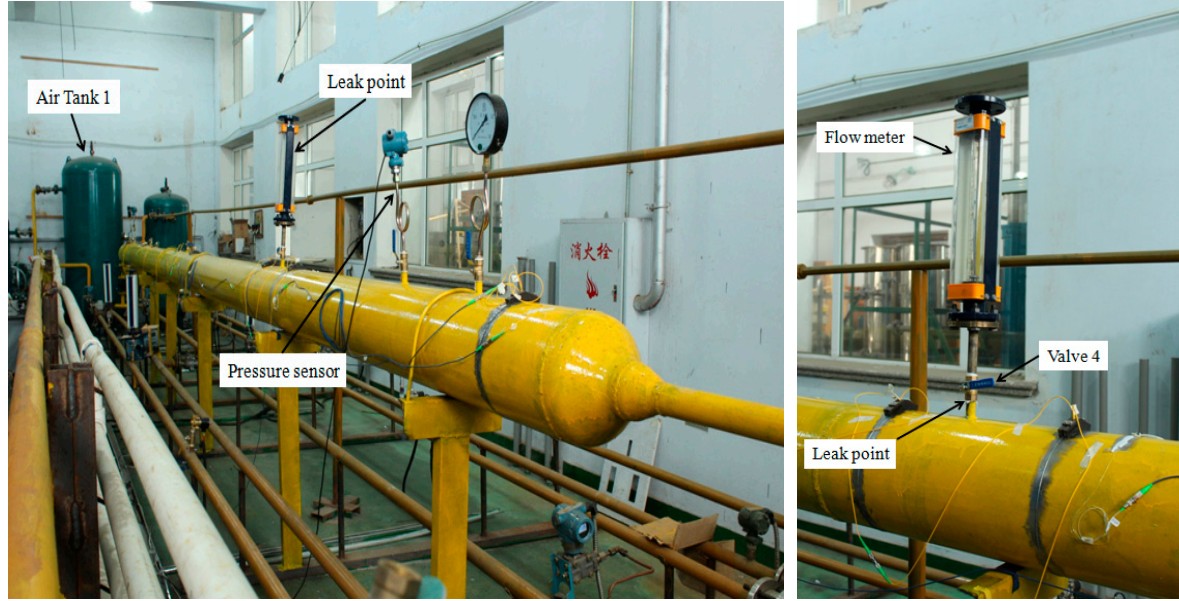

(**a**) Pipeline and air tank　　　　　　　　　　　(**b**) Leak point

**Figure 8.** Photos of the gas pipeline leak detection testbed.

After 20 s, V4 is closed to stop the leak.

The state vector is:

$$X_j = [P_{1,j}, P_{2,j}, P_{3,j}, \cdots, P_{31,j}, Q^-_{1,j}, Q^-_{2,j}, \cdots, Q^-_{31,j}, K_{2,j}, K_{3,j}, \cdots, K_{30,j}]^T.$$

The initial condition is:

$$\hat{X}_0 = [P_{1,0}, P_{2,0}, P_{3,0}, \cdots, P_{31,0}, Q^-_{1,0}, Q^-_{2,0}, \cdots, Q^-_{31,0}, K_{2,0}, K_{3,0}, \cdots, K_{30,0}]^T.$$

Among them,

$$P_{1,0} = 800 \text{ kPa}, P_{31,0} = 750 \text{ kPa}$$

$$Q^-_{1,0} = Q^-_{2,0} = \cdots = Q^-_{31,0} = 132 \text{ kg/h}$$

$$K_{2,0} = K_{2,0} = \cdots = K_{30,0} = 0.$$

There are three observation points of starting pressure, flow, and terminal pressure in this experiment. Therefore, the matrix $H$ has the dimension of $3 \times 91$, where $H(1,1) = 1$, $H(2,31) = 1$, $H(3,32) = 1$ and the remainder of elements were 0.

The measurement noise covariance was estimated by the data collected in the experiment. The setting of process noise was similar to the simulation example.

The computational steps in the simulation example were repeated to obtain the estimated leakage rate and leakage location with the EKF, as shown in Figures 9 and 10.

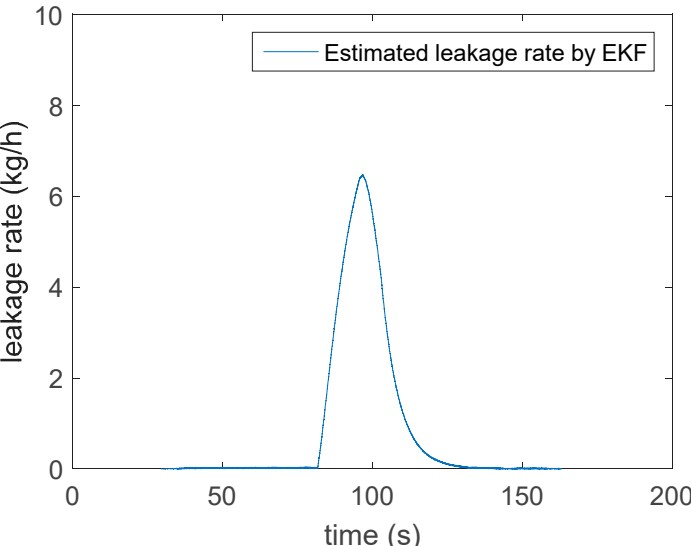

**Figure 9.** Estimated leakage rate using experimental data.

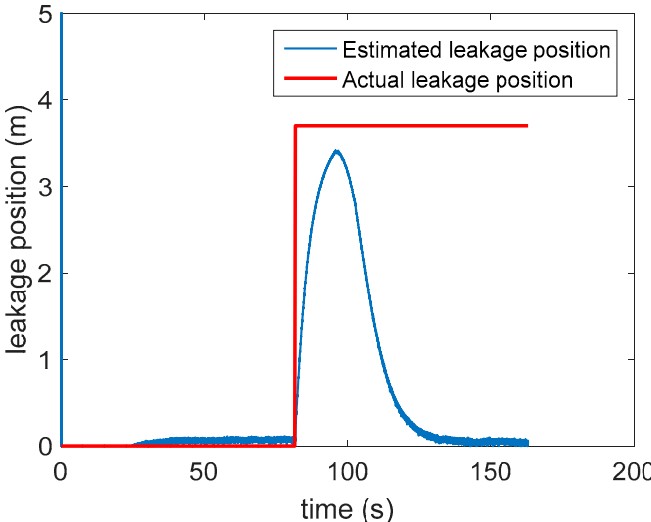

**Figure 10.** Estimated leak position using experimental data.

As can be seen from Figure 9, the estimated leakage rate increased between 80 and 100 s and then declined thereafter. This is consistent with the experimental procedure where leakage commences at 80 s through the opening of the ball valve. By 100 s, the leak has stopped and the leak was reduced to zero. It can be seen from the analysis that the change of the estimated leakage rate is consistent with the change of the actual leakage rate. Since the leakage state cannot be maintained under experimental conditions, the estimation of the leakage rate is affected. It can be assumed that if the leakage was not stopped during the experiment and the leakage was kept to a steady state, the estimated leakage rate of EKF would maintain a relatively stable state after rising to the highest point, which is similar to the result in Figure 6. In the experiment, the maximum value of the leakage rate read by the rotor flowmeter at the leakage point 1 was about 6.8 kg/h, and the highest point of the estimated curve of leakage rate was about 6.2 kg/h. The absolute error was 0.6 kg/h and the relative error was 8.8%. Experimental results and simulation results show that the method based on EKF is feasible to detect the leakage of a natural gas pipeline and estimate the leakage rate.

The red line in Figure 10 represents the actual leakage location, and the blue line represents the estimated curve of the leakage location. It is not difficult to explain why the estimated curve of the leakage position rises and then falls, according to the change of the estimated value of leakage

rate and Equation (13). The actual pipeline leakage point 1 is 3.9 m away from the pipeline starting point pressure sensor P1, while the peak of the estimated leakage position curve is about 3.3 m. The absolute error is 0.6 m. As the distance between pressure sensor P1 and P2 is 9 m, the relative error is 6.7%. It is conjectured that if the leakage was not stopped, the positioning accuracy would be further improved. The result of the experiment shows that the proposed method based on EKF is feasible to locate leakages in natural gas pipelines.

## 7. Conclusion and Future Work

This paper introduces a method of natural gas pipeline leak detection based on the extended Kalman filter (EKF). The change of flow state in an actual pipeline operation is usually random and easily affected by all kinds of interferences. Therefore, it is difficult to reflect the real-time state of flow by the transient flow real-time model method directly. This paper proposes a time-varying real-time EKF method for pipeline leak detection. In applying the approach to simulated and laboratory case studies, this paper has shown that the separate treatment of system and measurement noise inherent within EKF means that the proposed approach has low sensitivity to random noise, and good capability for accurately estimating leakage rate and location. The results of simulation and experiment show that the method is sensitive to small leakages of a pipeline and the estimation of leakage rate and location are accurate.

As the foundational work on an algorithm and experimental study, there are several limitations which we would like to address in future work. Firstly, the algorithm is only applied to long straight pipeline but not a pipeline network with complex topology. Secondly, the experimental study is verified on a lab testbed that is smaller than typical real-world gas pipelines. It will be critical to work with gas companies to experiment with real gas pipelines. Thirdly, while the probability of a multi-point leakage is much lower than that of a single-point leakage, it is still useful to conduct research in detecting leakage in more complex cases.

**Author Contributions:** Q.H. derived the algorithms, led the experiments and wrote the paper. W.Z. revised and proofread the paper.

**Funding:** This research was funded by Scientific Research Project of Harbin University of Commerce (No. 17XN013) entitled "Research about detection and location of natural gas pipeline leakages based on Kalman filter and Fiber sensing," the Doctoral Research Project of Harbin University of Commerce (No. 2016BS19), and National Natural Science Foundation of China (No. 51808092).

**Conflicts of Interest:** The authors declare no conflicts of interest.

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
