# Peer review of "An EKF-Based Method and Experimental Study for Small Leakage Detection and Location in Natural Gas Pipelines"

_applsci, doi:10.3390/app9153193_

Round 1

Reviewer 1 Report

This paper presented an EKF based natural gas pipeline leak detection and localization method. The extended Kalman filter is applied to the nonlinear transient model of the pipeline process through linearization. A simulated case study and a lab experiment are conducted to demonstrate the performance of the method.

The paper can be accepted for publication after clarifying the following issues:

1. Since EKF has been applied to pipeline leak detection in the literature, it is necessary to clarify what is the novelty of this work.

2. There are many practical issues for implementing the model-based EKF method. Uncertainties exist in various parameters such as pipe diameter, pipe roughness, friction coefficient, temperature change along the pipeline, the compressibility of the gas, etc. A discussion on how the proposed method can robustly handle (some of) the above uncertainties is needed.

3. Please discuss the observability of the process.

4. Is there any guideline on how to select the number of segments for the pipeline? How does this parameter affect the localization accuracy? 

Reviewer 2 Report

Gas leakage and location identification is an very important and interesting topic to study. The authors used extended Kalman filter (EKF) analysis to detect and locate small leaks in natural gas pipelines.  Both the simulation and experimental results shows the valid of the proposed method.  Overall, it is a solid paper.  Two comments:

The method relies on many pressure and flow measurements on the pipeline.  Will those measurement avaliable for the nature gas pipeline network.

In actual system, the distance between the sensors are much further than the testing set up. The time delay from the sensor to the controller will have large impacts on the accuracy, especially the leak location detection. 

Reviewer 3 Report

The paper presents a method based on EKF for small leakage detection in pipeline systems. The proposed method was verified on the simulation data and by the lab-scaled experiment. Overall, the paper is interersting but it need major modifications before it can be accepted for publication.

The proposed method is based on EKF, thus the authors should do more critical review on EKF-based methods and explain the advances of the proposed method against the previous ones.

There are many EKF-based leakge detection methods developed in previous studies. Therefore, the novelties and contributions of this paper should be clearly explained in the introduction part.

The paper is about 'small leakage detection' by EKF.  How to define 'small' and 'large' leakage? Also, the authors should explain how their EKF-based method can detect small leakage?

Only one leakage point is considered in the simulation and experimental studies. This is not enough to demonstrate the feasibility and practicality of the proposed method. The method should be verifed on the pipeline model with mutli leakage points and at different positions.

The limitation and future studies should be included in the conclusion part.

Round 2

Reviewer 3 Report

The revised manuscript is okay, but so many cited references from their group (University of Houston).
